# Quantum coherence and interference of a single moiré exciton in nano-fabricated twisted monolayer semiconductor heterobilayers

Haonan Wang[1], Heejun Kim[1], Duanfei Dong[1], Keisuke Shinokita [1], Kenji Watanabe [2], Takashi Taniguchi [3] & Kazunari Matsuda [1] ✉

The moiré potential serves as a periodic quantum confinement for optically generated excitons, creating spatially ordered zero-dimensional quantum systems. However, a broad emission spectrum resulting from inhomogeneity among moiré potentials hinders the investigation of their intrinsic properties. In this study, we demonstrated a method for the optical observation of quantum coherence and interference of a single moiré exciton in a twisted semiconducting heterobilayer beyond the diffraction limit of light. We observed a single and sharp photoluminescence peak from a single moiré exciton following nanofabrication. Our findings revealed the extended duration of quantum coherence in a single moiré exciton, persisting beyond 10 ps, and an accelerated decoherence process with increasing temperature and excitation power density. Moreover, quantum interference experiments revealed the coupling between moiré excitons in different moiré potential minima. The observed quantum coherence and interference of moiré exciton will facilitate potential applications of moiré quantum systems in quantum technologies.

A quantum two-level system has garnered considerable attention in recent years due to its numerous potential applications in the fields of physics, such as quantum simulation, quantum computing, and quantum information processing[1–5]. The development of these systems facilitates the construction and utilization of quantum bits (qubits), which serve as fundamental units for quantum computing and quantum information[6–8]. Resonant light–matter interactions, such as Rabi oscillation[9,10], Ramsey interference[11], and Hahn echoes[12], enable the manipulation of quantum two-level systems[13,14] by generating superposition states. However, the superposition states of qubits suffer from the interaction and fluctuations from the environment, resulting in an accelerated decoherence process[15]. This decoherence process imposes a temporal limitation on the precise manipulation of quantum systems[16–18], hindering their potential applications[19,20]. Hence, for platforms aiming to achieve qubits, a sufficiently long coherence time is imperative[21,22]. Furthermore, it is also necessary to control the interaction between quantum systems, since such interaction not only modifies the quantum coherence within each individual system but also facilitates the formation of a coupled-quantum system. This introduces interference or entanglement[23] between systems, which is essential for the development of large-scale quantum devices[24–26].

Recent progress in artificial van der Waals (vdW) structures, achieved by stacking atomically thin two-dimensional (2D) materials,

[1]Institute of Advanced Energy, Kyoto University, Uji, Kyoto 611-0011, Japan. [2]Research Center for Electronic and Optical Materials, National Institute for Materials Science, 1-1 Namiki, Tsukuba, Ibaraki 305-0044, Japan. [3]Research Center for Materials Nanoarchitectonics, National Institute for Materials Science, 1-1 Namiki, Tsukuba, Ibaraki 305-0044, Japan. ✉e-mail: matsuda@iae.kyoto-u.ac.jp

has opened up opportunities for designing novel quantum platforms[27–30]. vdW heterobilayer assembled from monolayers of semiconducting transition metal dichalcogenides exhibited various intriguing physical phenomena, including strongly correlated insulator phases[31,32], superconductivity[33], and novel ferromagnetism[34]. Moiré superlattices with varying atomic registries in vdW heterobilayers can be constructed using monolayer semiconducting transition metal dichalcogenides with a small lattice mismatch or twist angle. The resulting moiré superlattice leads to the formation of periodic, ordered potential traps, confining and spatially organizing optically generated bound electron–hole pairs (excitons) into periodic arrays of quantum two-level systems[35–38]. The trapped excitons in moiré potentials are expected to exhibit long quantum coherence due to their limited degree of freedom as 0D quantum systems[39,40]; furthermore, coupling interactions can be formed between spatially separated moiré potentials, leading to quantum interference of emitted photons[41]. These results establish moiré exciton quantum systems as not only a promising platform for achieving extended coherence but also an effective tool for exploring interactions within or between quantum systems. However, experimentally, important information on the quantum coherence and interference of moiré excitons in vdW heterobilayers remains unexplored. This is due to the overlapping of multiple emission peaks from moiré excitons in their inhomogeneously broadened spectra, hindering these intrinsic insights within the diffraction limit of light. The experimental approaches of strain-induced exciton traps introduced by metallic nanopillars in heterobilayers have been previously reported, to observe spectra with discrete emission lines[42,43]; however, the intrinsic properties of the moiré exciton system might be inadvertently concealed due to the additional strain-induced effects.

In this study, we provide a approach of quantum optics in semiconducting twisted heterobilayer beyond the diffraction limit of light. A single and sharp optical signal from a single exciton trapped in the moiré superlattice was successfully demonstrated in MoSe$_2$/WSe$_2$ heterobilayer, enabling the direct measurement of the first-order correlation function of moiré exciton emission. The quantum coherence of a single moiré exciton within a potential was maintained for over 12 ps at a low temperature of 4 K, considerably longer than the coherence of a 2D exciton in a monolayer semiconductor[44–46]. Quantum beats were experimentally observed with a period of approximately 230 fs, indicating the coupling between moiré excitons trapped in different potentials. Furthermore, this study discussed the mechanism of quantum decoherence of a single moiré exciton state with increasing temperature and excitation power mediated by the moiré exciton–phonon and exciton–exciton interaction.

## Results and discussion

Figure 1a and b illustrate schematics of the concept using a nanostructure fabrication process, and experimental setup employed in this study. Nanoscale fabrication using reactive-ion etching (RIE) enables emissions from a single moiré exciton and the observation of its quantum coherence in the MoSe$_2$/WSe$_2$ heterobilayer. During usual optical measurements, obtaining clear spectrum from moiré excitonic states is difficult due to the inhomogeneity of the moiré potential, leading to ensemble-averaged and broadened emissions comprising multiple peaks. This issue arises because the focused laser light with a spot diameter of approximately 1.5 μm is determined by the diffraction limit of light, which excites a large number of moiré potentials due to the much smaller spatial period of moiré potentials. To address this, in the microfabrication process, we applied a nanoscale fabrication technique to reduce the optical excitation and detection area of the MoSe$_2$/WSe$_2$ heterobilayer with a nano-pillar structure. The nanofabricated heterobilayer, with a pillar size smaller than the wavelength of light, will result in a reduced number of spectral peaks beyond the diffraction limit of light. Thus, this approach allows for the anticipated

observation of emission from a single moiré potential, enabling the revelation of the quantum coherence of moiré excitons.

The details of the RIE procedure for preparing the nanostructure are described in the Methods section[47]. Figure 1a displays the designed pattern for nanostructure fabrication. Black circular areas with a diameter of 2 μm, larger than the focused laser spot size, are exposed to reactive ions and subsequently etched. The inner white areas are shielded during the RIE process via electron beam resist, and the resultant pillar sizes (D) corresponding to the optically active size of the heterobilayer are designed to be 50, 100, 150, 200, and 500 nm. Figure 1b presents a typical scanning electron microscopy (SEM) image of the fabricated nanostructure with an actual pillar size of 240 nm in the heterobilayer. Dotted circle at the center corresponds to the pillar region in Fig. 1a, while the area between the outer and inner dotted circles denotes the etched region. The SEM images of various pillar sizes are presented in Supplementary Fig. 3.

Figure 1c displays the photoluminescence (PL) spectra of the nanostructure-fabricated MoSe$_2$/WSe$_2$ heterobilayer, measured at 4 K with varying pillar sizes. In the heterobilayer with a pillar size (D) of 500 nm, the PL spectrum reveals an inhomogeneously broadened ensemble average of multiple peaks from large number of moiré excitons. This finding agrees with previously reported results[48]. The broadened spectrum, characterized by a Gaussian distribution, arises from the inhomogeneity of moiré potentials in the heterobilayer. As the pillar size decreases, the number of peaks in the spectra significantly decreases. Consequently, the PL spectrum of the heterobilayer with a 50 nm pillar size reveals a singular emission peak from a moiré exciton, attributed to the reduced number of moiré potentials within the optical excitation and detection area determined by the pillar size. An additional series of PL spectra for different pillars in the heterobilayer are shown in Supplementary Fig. 2a, and the results are similar to those in Fig. 1c.

Figure 1d shows the integrated PL intensity from moiré exciton as a function of the actual pillar size. As the pillar size decreases, the integrated intensity is reduced rapidly, particularly smaller sizes. This reduction is attributed to the Gaussian distribution of the excitation laser profile, which results in the varying power densities across different pillar sizes, as presented in Supplementary Fig. 2b. The integrated intensities are calibrated against the average laser intensity. The results demonstrate a linear decrease in intensity with pillar size, which strongly supports the reduction of the number of optically excited and detected moiré potentials within the nanostructure-fabricated MoSe$_2$/WSe$_2$ heterobilayer (Supplementary Fig. 2c).

### Power dependence of PL spectra

Figure 2a presents a contour plot showing the excitation power-dependence of the normalized PL spectra in the range of 0.8–3124 W/cm$^2$ in the MoSe$_2$/WSe$_2$ heterobilayer with D = 50 nm at 4 K. The power densities are calibrated based on the pillar size and laser spot size, as previously described. The spectral shape of the PL spectra varies with the excitation power density. Figure 2b displays the PL spectra of a moiré exciton normalized to the excitation power density. At a low excitation power density of approximately 0.8 W/cm$^2$, a sharp peak from a moiré exciton is observed at 1.380 eV with a linewidth of 600 μeV, as defined by the spectral resolution in this setup condition. As the excitation power density increases, the normalized PL peak of a moiré exciton at 1.380 eV gradually decreases, and additional spectral peaks emerge.

Figure 2c illustrates the excitation power dependence of the PL spectra of a moiré exciton on an expanded energy scale to clearly see changes in the spectra. At low excitation power densities, specifically below 7.8 W/cm$^2$, a single PL peak is observed at 1.380 eV because the nanostructure fabrication limits the observed number of moiré potentials. As the excitation power density increases, the primary PL peak at 1.380 eV begins to broaden and exhibits saturation behavior

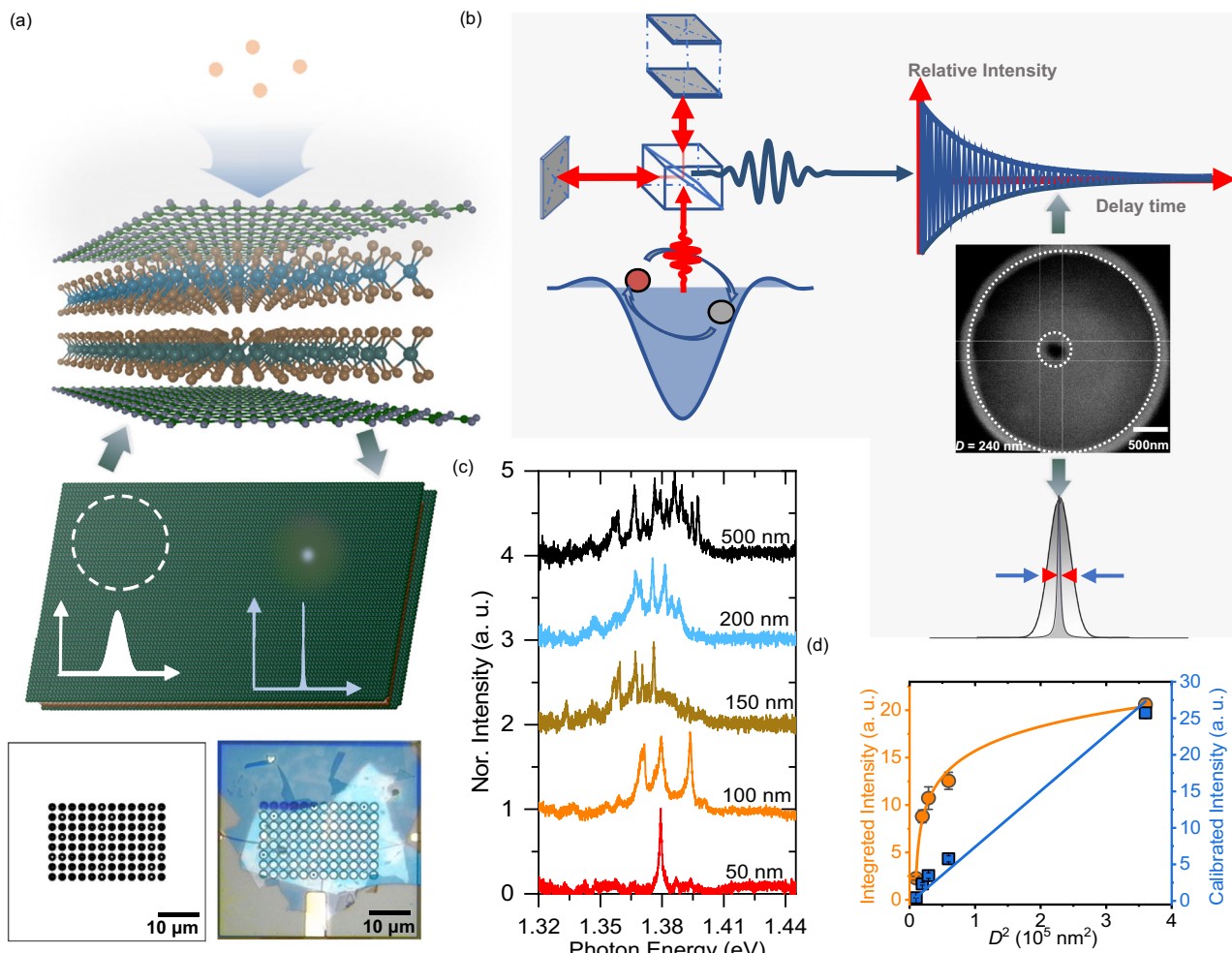

**Fig. 1 | Nanoscale fabrication. a, b** Schematic of the concept in this study. Nanoscale fabrication using reactive-ion etching enables to obtain emission from a single moiré exciton and the observation of its quantum coherence with Michelson interferometer in MoSe₂/WSe₂ heterobilayer beyond the diffraction limit of light. Designed pattern for RIE (left) and the optical image of MoSe₂/WSe₂ heterobilayer with an array of nanofabricated structures (right) are presented in (**a**). The designed sizes of pillar used in the nanostructure fabrications are 50, 100, 150, 200, and 500 nm. A SEM image of typical pillar is presented in (**b**). The inner dotted circle shows a pillar with a diameter of 240 nm corresponding to the optical excitation and observation area of moiré potential. Optical spectra from the pillars are Fourier-transformed to temporal interferograms using the Michelson interferometer. **c** PL spectra of MoSe₂/WSe₂ heterobilayers with various pillar sizes at 4 K. **d** Integrated PL intensity and calibrated intensity of heterobilayer as a function of pillar size. Solid line in the image represents the guide line, wherein calibrated intensity is determined as integrated intensity/average laser power density. The calibrated intensity shows linearly dependence on $D^2$, indicating the linearly dependence of peak numbers on the pillar size.

around 15.6 W/cm², which will be discussed later. Moreover, above 15.6 W/cm², an additional PL peak appears at the higher energy side of primary PL peak at 1.382 eV.

In order to assign the origin of spectral peaks, the PL intensity of the peak as a function of excitation power density is plotted in Fig. 2e. The PL intensity at 1.380 eV increases linearly and then gradually saturates at approximately 15.6 W/cm², consistent with the spectral behaviors observed in Fig. 2d. The saturation power density corresponds to a generated exciton density of $9 \times 10^{11}$ cm⁻², which is almost consistent with the estimated moiré potential density of $3 \times 10^{12}$ cm⁻² in the MoSe₂/WSe₂ heterobilayer with a twist angle of $56.5° \pm 0.3$, as indicated by the SHG results shown in Supplementary Fig. 1b. Moreover, the circularly and linearly polarized PL spectra are measured to further support the moiré exciton emission, as shown in Supplementary Fig. 8 and Supplementary Fig. 9. The polar pattern of PL intensities detected through linear polarizers as a function of the detection angle indicates that the emission state maintains $C_{3v}$ symmetry, which is expected for a moiré exciton system[49]. Circular polarization–resolved spectroscopy shows stronger co-polarized PL intensities, which is also

consistent with the results of the H-type stacking of the heterobilayer[29]. The above experimental results support the finding that the primary PL peak at 1.380 eV originates from the recombination of moiré exciton. The PL peak at 1.382 eV emerges after the saturation of moiré exciton emission above 15.6 W/cm², and the nonlinear increase of PL intensity shows a square dependence on the excitation power density in Fig. 2e, which suggests that the PL peak is attributed to the recombination of moiré biexciton confined within the potential. Moreover, the blueshifted emission of moiré biexciton than that of moiré exciton due to dipolar repulsion between excitons and the energy difference of 2 meV corresponding to the binding energy of moiré biexciton are consistent with previously reported results[50].

Figure 2d displays the spectral linewidth of moiré exciton peak as a function of the excitation power density, derived from the Voigt function fitting procedure. The spectral linewidth exhibits a narrow value of 600 µeV at lower excitation conditions below 10 W/cm², where the PL intensity increases linearly. Above a saturation power density of 15.6 W/cm², the spectral linewidth of the moiré exciton peak becomes increasing dependent on the excitation power density,

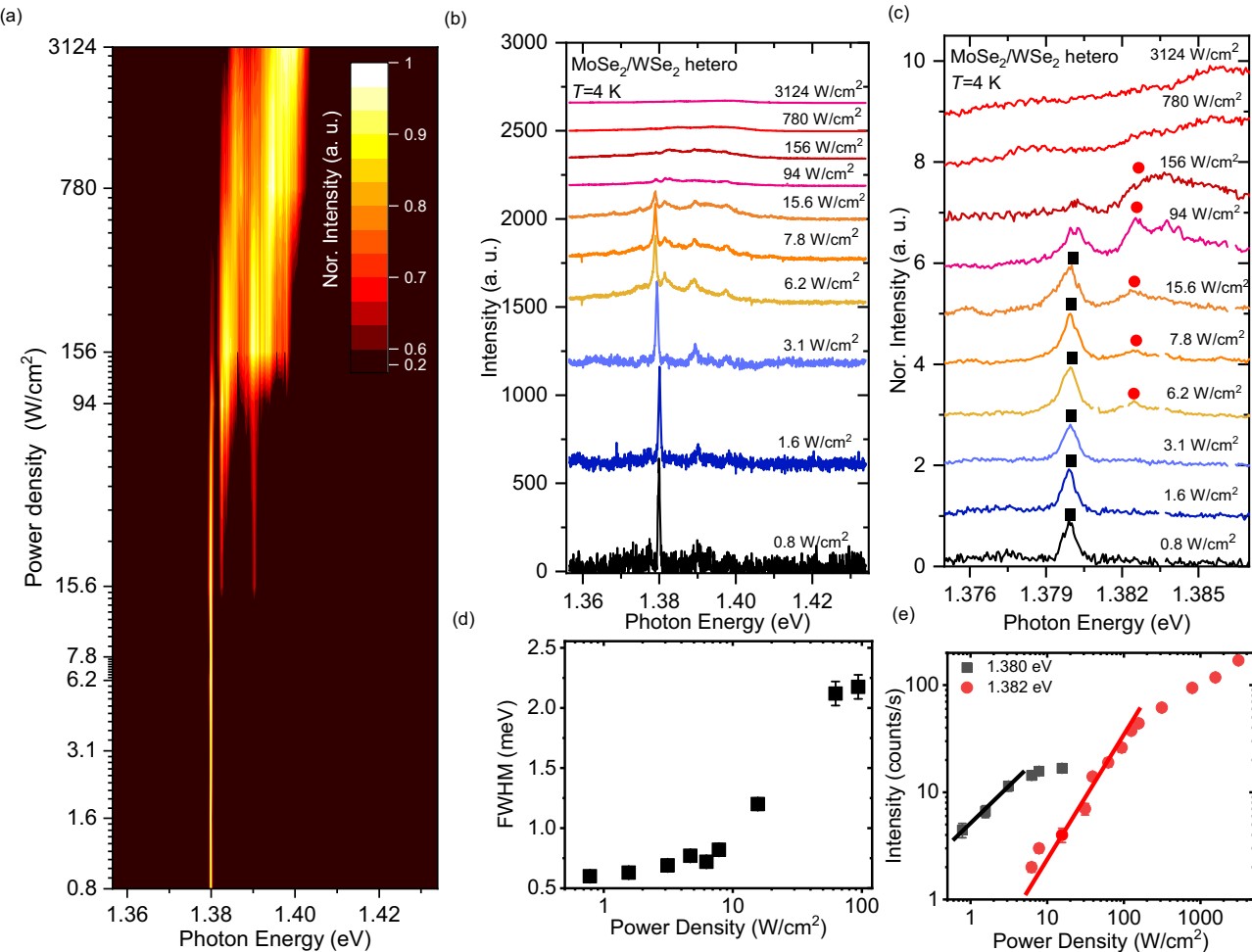

**Fig. 2 | Power dependence of PL spectra. a** Contour plot of the normalized PL spectra of MoSe₂/WSe₂ heterobilayer at 4 K for various discrete excitation power densities ranging from 0.8 W/cm² to 3124 W/cm². **b** Low-temperature PL spectra of heterobilayer with a pillar diameter of 50 nm for different excitation power densities. **c** PL spectra of heterobilayer in the expanded energy scale. **d** Spectral linewidth of PL peak at 1.380 eV represented by full width at half maximum (FWHM) as a function of excitation power densities. **e** PL intensities of peaks at 1.380 and 1.382 eV as a function of excitation power density. Red and black lines denote the linear and square excitation power dependence.

which suggest the influences of exciton density on the moiré exciton coherence.

## PL spectral wandering and first-order correlation function

To quantify intrinsic spectral broadening, we measured the time-dependent PL spectra in the MoSe₂/WSe₂ heterobilayer. Figure 3a presents the time evolution of the moiré exciton PL spectra, with each spectrum accumulated over 30 s. Random fluctuations in spectral peak positions, known as, spectral wandering or spectral jittering, are clearly observed in the moiré exciton emission, which is a characteristic of 0D quantum systems[51]. Figure 3b traces the energy peak positions derived from the data in Fig. 3a. The frequency distribution of each peak position is shown as a histogram in Fig. 3c. The histogram reveals that the energy positions during spectral wandering range from 1.3755 to 1.3758 eV. The spectral wandering of the PL peak is also observed across different pillars of the heterobilayer (refer to Supplementary Fig. 10 and Supplementary Fig. 16).

To obtain the information on the quantum coherence, the first-order correlation function $g^{(1)}(\tau)$ of emission signals from a moiré exciton is measured using a Michelson interferometer (Supplementary Fig. 11). Figure 3d presents the contour map of interferometry of the PL spectra as a function of delay time at 4 K for an excitation power density of 14 W/cm². The amplitude of the oscillation fringe between

the maximum and minimum intensities gradually decreases with increasing delay time, indicating the process of decoherence, as presented by the temporal interferogram in Fig. 3e. The visibility $V(\tau)$ is calculated as follows:

$$V(\tau) = \frac{I_{\max} - I_{\min}}{I_{\max} + I_{\min}} \quad (1)$$

where $I_{\max}$ and $I_{\min}$ denote the maxima and minima intensities obtained from an oscillation period around a certain delay time in the interferometry. The visibility $V(\tau)$ (blue circle) as a function of the delay time is plotted in Fig. 3f. The visibility as a function of delay time, corresponds to the Fourier transform of the emission spectrum, with the convolution result of extrinsic inhomogeneous and intrinsic inhomogeneous linewidth, in form of Lorentz and Gaussian functions, respectively. Consequently, the delay time-dependent visibility in Fig. 3f can be modeled by the product of exponential and Gaussian functions, as follows[52,53]:

$$V(\tau) = e^{-B_1\tau} \cdot e^{-B_2^2\tau^2} \quad (2)$$

where $B_1$ and $B_2$ are parameters defined in the exponential and Gaussian functions, respectively. The fitted result using Eq. (2) with $B_1$

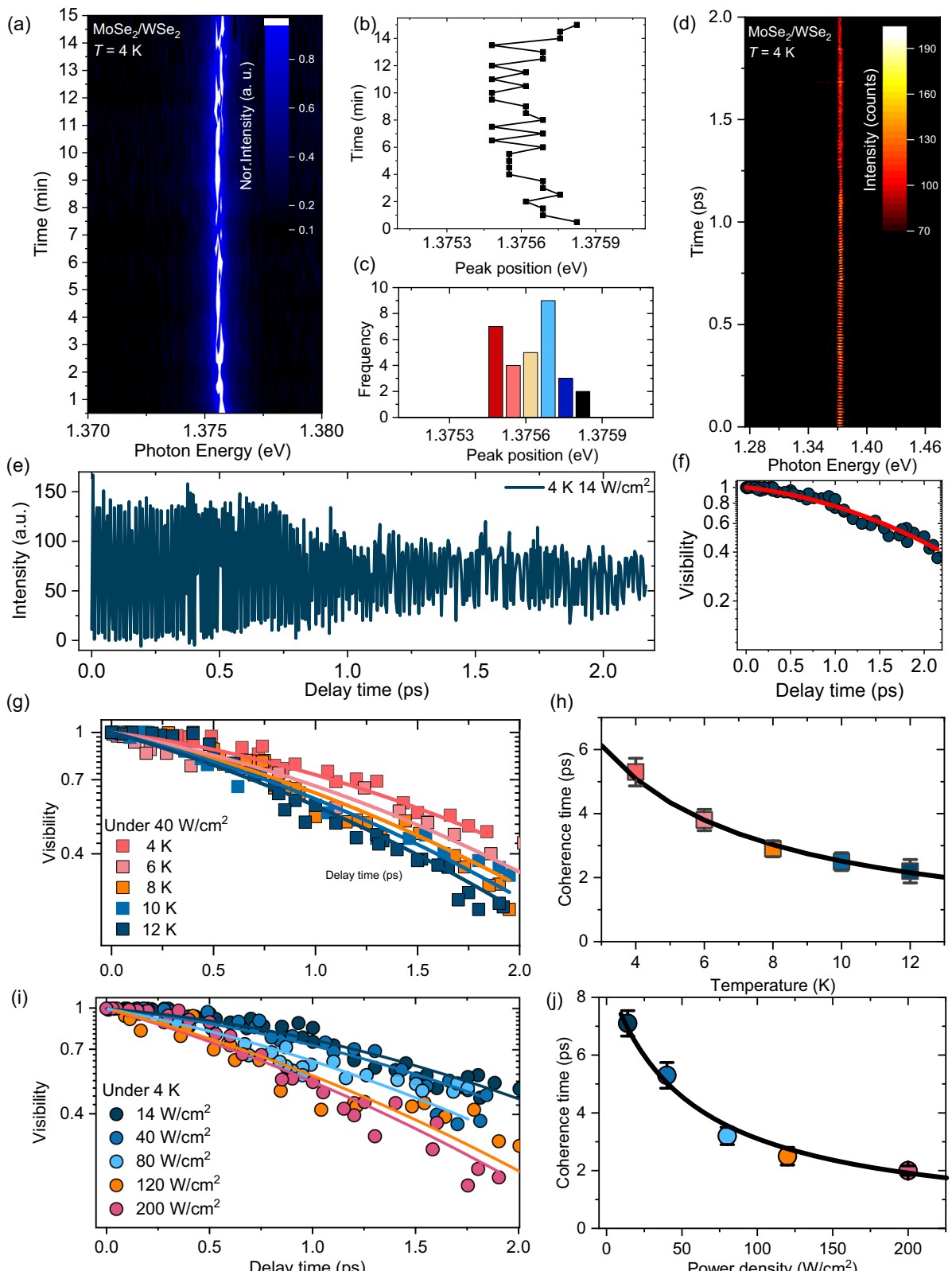

**Fig. 3 | PL spectral wandering and first-order correlation function. a** Time evolution of PL spectrum of moiré exciton peak at low temperature, with each spectrum accumulated for 30 s. **b** Time-trace of energy peak position for each spectrum derived from the contour plot. **c** Frequencies of energy peak positions represented as a histogram. **d** Contour plot of the first-order correlation function of PL signals as a function of delay time, measured using the Michelson interferometer. **e** Interferogram of the moiré exciton peak in the time domain at 4 K in the excitation power condition of 14 W/cm². **f** Decay profile of visibility in the interferogram, with solid curve representing the fitted result of the product of an exponential and a Gaussian function. **g** Temperature dependence of the visibility of the moiré exciton peak as a function of delay time. Solid curves indicate the fitting results of the product of an exponential and Gaussian function. **h** Plots of the extracted coherence time of moiré exciton ($T_2$) as a function of temperature. **i** Visibility of the moiré exciton peak as a function of delay time for different excitation power densities. **j** Plot of coherence time as a function of power density.

( $= 0.14\,\text{ps}^{-1}$) and $B_2$ ( $= 0.35\,\text{ps}^{-1}$) well reproduce the experimental results of visibility as a function of delay time. In the spectral domain, the homogeneous and inhomogeneous broadening of the spectrum are described as $2\hbar B_1$, and $4\hbar\sqrt{\ln 2}B_2$, where $\hbar$ is the Planck constant divided by $2\pi$.

The exponential decay rate of $B_1$, in the visibility of interferometry is inversely related to the coherence time of the quantum state as $T_2 = \frac{1}{B_1}$. The coherence time ($T_2$) of a quantum state is directly determined by the exciton lifetime and pure dephasing time, as described by the following equation

$$\frac{1}{T_2} = \frac{1}{2T_1} + \frac{1}{T_2^*} \qquad (3)$$

where $T_1$ represents the energy relaxation lifetime and $T_2^*$ denotes the pure dephasing time. The lifetime of the interlayer moiré exciton in $MoSe_2/WSe_2$ heterobilayer was measured using time-resolved PL spectroscopy, employing a time-correlated single-photon counting method (Supplementary Fig. 13a) from 4 to 14 K under the excitation power density of $56\,\text{W/cm}^2$. The PL decay profiles of moiré exciton exhibit a longer decay time of several tenths of ns, which is consistent with previously reported results[48]. The decay curves are modeled by exponential functions as $I(t) = A_1\exp(-t/\tau_1) + A_2\exp(-t/\tau_2) + A_3\exp(-t/\tau_3)$, where $A_1$, $A_2$, and $A_3$ as well as $\tau_1$, $\tau_2$, and $\tau_3$ are the coefficients of amplitude and decay times. The fitting results yield the values of $\tau_1 = 14\,\text{ns}$, $\tau_2 = 100\,\text{ns}$, and $\tau_3 = 700\,\text{ns}$ at 4 K. As temperature increases, the PL decay profiles become faster, and the obtained decay times are summarized in Supplementary Fig. 13b. The $\tau_2$ can be ascribed to the decay lifetime of moiré exciton[54]. Given that $\tau_2$ is much longer compared to the measured coherence time $T_2$, the population relaxation process hardly contributes to the dephasing process. As a result, pure dephasing time of this position is evaluated to be 7.1 ps, corresponding to a homogeneous linewidth of 184 μeV.

Supplementary Fig. 15 exhibits the interferograms of moiré exciton emission at various temperatures from 4 to 12 K. The decay of interferograms is progressively faster with increasing temperature. Figure 3g presents the temperature dependence of visibility, which are also well fitted using Eq. (2) by altering the value of $B_1$, as shown in Fig. 3h. The evaluated coherence times of moiré exciton considerably decreases from 5.3 to 2.2 ps, corresponding to an increase of the homogeneous linewidth from 250 μeV to 600 μeV. The broadening of the intrinsic homogeneous linewidth with increasing temperature affects the PL linewidth, which is consistent with the results in Supplementary Fig. 5. The broadening of the homogeneous linewidth with increasing temperature can be linearly modeled using $\Gamma(T) = \gamma_0 + \gamma' T$, where $\gamma_0$ is the residual homogeneous linewidth at zero temperature. The coherence time $T_2$ can be described by $T_2 = 2\hbar/(\gamma_0 + \gamma' T)$ due to the relationship of $\Gamma_{\text{homo}} = 2\hbar/T_2$. Solid line in Fig. 3h reproduces the experimental results at various temperatures. According to the fitting result, the value of the residual homogeneous linewidth is 83 μeV, corresponding to a coherence time of 16 ps at the zero-temperature limit, and the linear coefficient of $\gamma'$ as a function of temperature is 43 μeV/T. The temperature-dependent linear increase in homogeneous linewidth implies that the decoherence of moiré exciton is determined by the interaction of low-energy acoustic-phonon modes in the heterobilayer. Moreover, the value of the broadening coefficient $\gamma'$ dominated by the strength of exciton–acoustic phonon interactions, is smaller than that of 2D exciton (60 μeV/T) in a monolayer semiconductor[55]. This result implies that the exciton-phonon interaction is suppressed by the quantum confinement of moiré potential.

Supplementary Fig. 14 presents the interferograms of moiré exciton emission at 4 K for different excitation power densities. With increasing excitation power density, the decay of the interferograms increases. Figure 3i presents the power density dependence of visibility, which also fits well using Eq. (2) with changing the value of $B_1$, as

shown in Fig. 3j. The evaluated coherence times of moiré exciton depending on the excitation power density significantly decrease from 7.1 to 2.0 ps, corresponding to a homogeneous linewidth increasing from 180 to 660 μeV. The broadening of homogeneous linewidth with increasing excitation power density were linearly fitted by $\Gamma(P) = \beta_0 + \beta P$, where $\beta_0$ is the homogeneous linewidth under weak excitation power limit. Thus, the coherence time can be modeled by $T_2 = 2\hbar/(\beta_0 + \beta' T)$, represented by the black solid line in Fig. 3j, which indicates a residual homogeneous linewidth of 160 μeV and coherence time of 8.2 ps under zero exciton density. The value of the broadening coefficient $\beta'$ resulting from the exciton–exciton interaction is estimated to be $2.6 \times 10^{-3}\,\text{meVcm}^2/\text{W}$, significantly lower than that previously reported for monolayer $WSe_2$[55]. The reduced coefficient suggests that moiré potential confinement, substantially decreases the interactions between excitons[56].

## Quantum beat and coupling between two moiré excitons

We further investigated the quantum interference of moiré excitons from the first-order correlation function of the corresponding signals in another pillar of the heterobilayer. Figure 4a shows the results of the PL spectra of moiré excitons for various excitation power densities. At the lowest excitation power ($0.8\,\text{W/cm}^2$), two peaks are observed at 1.327 and 1.309 eV, respectively, which is indicated by the solid circle and rectangle. With increasing excitation power density, each peak shows saturation behavior independently, and new peaks appear at the higher energy side. Intensities of these two peaks are plotted as a function of the excitation power density in Fig. 4b. The intensities of each peak are fitted with function $I = P^\alpha$, represented by the solid lines. These two peaks exhibit linear increases at low excitation power densities and reach saturation at different power densities, indicating interlayer excitons trapped in moiré potentials of different depths ($M_1$ and $M_2$). The peak at 1.309 eV is associated with a deeper moiré potential ($M_2$) compared to the peak at 1.327 eV ($M_1$), which is caused by inhomogeneity within the optical excitation area.

Figure 4c displays a contour plot of the first-order correlation function of the moiré exciton PL signals measured at an excitation power density of $3.2\,\text{W/cm}^2$. Figure 4d demonstrates the interferogram of the peak at 1.327 eV, where the integrated peak intensity ($M_1$) is plotted against delay time. The envelope of the interferogram as a function of delay time presented in Fig. 4e shows a coherence time of 12 ps, corresponding to a homogeneous linewidth of 110 μeV. In contrast to the previous results, the interferogram in Fig. 4d presents a distinct beating pattern comprising multiple periods. The beating period $B_M$ is estimated to be $230 \pm 10$ fs, corresponding to an energy splitting of $18.0 \pm 0.8$ meV. According to the spectra in Fig. 4a, the peak positions of $M_1$ and $M_2$ show an energy difference of 18 meV, which is well matched to the splitting energy evaluated from the beating periods observed in the time domain. Because $M_1$ and $M_2$ are moiré excitons trapped in different moiré potential minima, it can be confirmed that the beating signal comes from the coupling of moiré excitons ($M_1$ and $M_2$). As indicated in Fig. 4f, the coupling between moiré excitons ($M_1$ and $M_2$) creates a four-level system, including the coherently coupled state $M_{12}$, two excited states ($M_1$ and $M_2$) trapped in moiré potentials, and the shared ground state (G). The emissions from the coupled state lead to quantum interference (quantum beat) in the $M_1$ interferogram. Quantum couplings across adjacent electronic systems have been reported in quantum wells[57], colloidal quantum dots[58], and molecules[59,60]. However, we demonstrate new features of coupled-quantum systems based on moiré excitons in a semiconducting heterobilayer, which can be extended to manipulate multiple quantum states in periodic moiré superlattices.

In conclusion, we demonstrated a new nanofabrication strategy based on RIE for investigating quantum physics in a semiconducting twisted $MoSe_2/WSe_2$ heterobilayer. A reduction in the number of moiré potentials in the optical excitation and detection beyond the

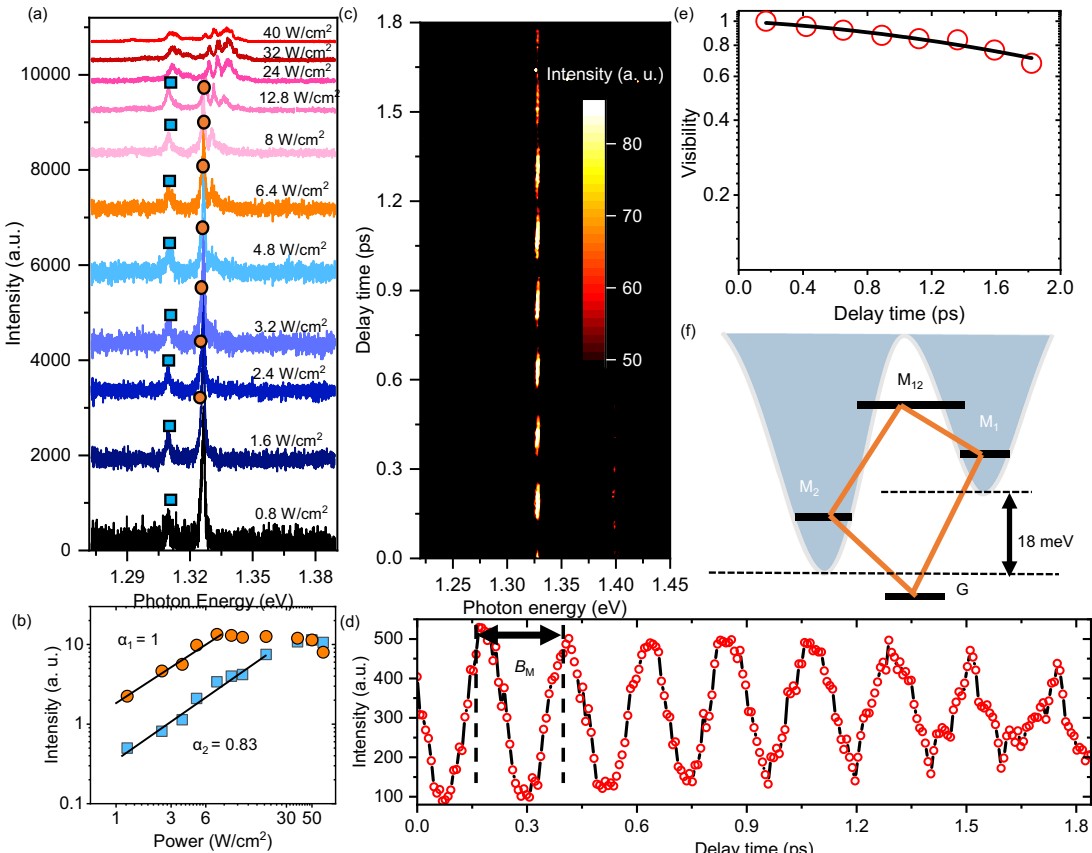

**Fig. 4 | Quantum beat and coupling between two moiré excitons. a** Low-temperature PL spectra (4 K) of a newly fabricated pillar with diameter of 50 nm for varying excitation powers. **b** PL intensities of peaks at 1.327 and 1.309 eV as a function of excitation power density. **c** Contour plot of the first-order correlation function of PL signals obtained using the Michelson interferometer as a function of delay time at the same position under excitation power of 3.2 W/cm². **d** Interferogram of the peak at 1.327 eV, displaying a quantum beat signal with a period $B_M = 230 \pm 10$ fs. **e** Envelope of the interferogram and its fitting result. **f** Schematic of the coupling of moiré excitons from two different moiré potential minima. $M_1$ indicates the moiré exciton state at 1.327 eV, while $M_2$ indicates the moiré exciton state at 1.309 eV. The period of beating signal reveals the energy splitting between two moiré excitons.

diffraction limit of light was realized by this nanofabrication, enabling the observation of optical signals from a single moiré exciton in the trapped potential. A significant single and sharp PL peak from a single moiré exciton in a potential has been successfully demonstrated, which also shows the characteristic spectral wandering in the 0D quantum system. This study explored the quantum coherence of a single moiré exciton for different temperatures and excitation power densities, confirming that the acceleration of decoherence was due to interaction of moiré exciton–acoustic phonon and moiré exciton–moiré exciton. Further, the long duration of quantum coherence observed in a single moiré exciton was revealed to be more than 12 ps at a low temperature of 4 K, which is much longer than that of an exciton in a monolayer semiconductor. Furthermore, quantum beats were observed in the interferogram of moiré exciton, proving the existence of coupling between moiré excitons trapped in different moiré potentials. The long coherence of moiré exciton revealed in this study offer potential applications of moiré quantum systems in quantum technologies.

## Methods
### Sample preparation and nanofabrication
Monolayer (1 L) MoSe₂, WSe₂, and encapsulating $h$-BN layers were prepared on SiO₂/Si substrates via mechanical exfoliation of bulk crystals. The layer number and thickness of MoSe₂ and WSe₂ were obtained from optical images and PL spectra. The MoSe₂/WSe₂ heterobilayer encapsulated by top and bottom $h$-BN was fabricated by a

polydimethylsiloxane (PDMS)-based dry-transfer method. The top $h$-BN, monolayer WSe₂, and MoSe₂ were sequentially picked up by poly (methyl methacrylate) (PMMA)-coated PDMS stamp and dropped onto the bottom $h$-BN on a SiO₂/Si substrate. The entire dry-transfer process was performed in an N₂-filled glove box. The fabricated sample was immersed after in acetone solution for removing residual PMMA. Electron beam lithography and selective reactive ion etching (RIE) using Ar gas were used for the fabrication of designed nanostructures in MoSe₂/WSe₂ heterobilayer–encapsulating $h$-BN layers. The selective RIE using Ar gas was performed in the conditions of power (50 W) and flow rate (40 s.c.c.m).

### Optical measurement
A linearly polarized semiconductor laser (2.38 eV) was employed as the excitation source for low-temperature PL measurements. A 50× objective lens with a numerical aperture of 0.67 was used for solely focusing on the excitation laser light on the surface and acquire optical images. The sample was positioned in a cryogen-free cryostat with a temperature ranging from 4 K to room temperature. The emission signals were coupled with an optical fiber and detected using a spectrometer and charge coupled device with a typical spectral resolution of 0.6 meV. A pulsed supercontinuum light source passed through a bandpass filter with a photon energy of 1.7 eV, with a repetition rate of 1 MHz was used for the time-resolved PL measurement. The emission signals were transported through the optical fiber, and detection was realized via an Si avalanche photodiode using a time-correlated single-

photon counting technique. A Michelson interferometer was used to gather the first-order correlation function $g^1(\tau)$ of the PL signals for coherence measurement. The detailed optical setup is shown in Supplementary Fig. 11.

## Data availability

Data presented in this paper and the supplementary materials are available from the corresponding author upon request.

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

## Acknowledgements

This work was supported by JSPS KAKENHI (Grant Numbers JP16H00910, JP16H06331, JP17H06786, JP19K14633, JP19K22142, JP20H05664, JP21H05232, JP21H05235, JP21H01012, JP21H05233, and JP22K18986), JST FOREST program (Grant Number JPMJFR213K), the Keihanshin Consortium for Fostering the Next Generation of Global Leaders in Research (K-CONNEX) established by the Human Resource Development Program for Science and Technology, MEXT, and the Collaboration Program of the Laboratory for Complex Energy Processes, Institute of Advanced Energy, Kyoto University. *h*-BN growth was supported by JSPS KAKENHI, (Grant Number JP20H00354) and by MEXT (Grant Number JPMXP0112101001). We thank Mr. Yuki Okamura, Dr. Yukiko Yamada-Takamura, Dr. Kohei Aso, and Dr. Yoshifumi Oshima for the help of TEM measurement, which was supported by Advanced Research Infrastructure for Materials and Nanotechnology in Japan (ARIM) of the Ministry of Education, Culture, Sports, Science and Technology (MEXT), Proposal Number JPMXP1222JI0020.

## Author contributions

H.W., K.W., and T.T. contributed to the fabrication of samples studied in this work. H.W., K.S., and K.M. designed the experiments, which were performed by H.W., H.K., and K.M. Data analysis was performed by H.W. and K.M. The draft was written by H.W., K.S., and K.M., with all authors contributing to reviewing and editing. The project was supervised by K.M.

## Competing interests

The authors declare no competing interests.
