## [Peer Review File · Nature Communications]

Quantum coherence and interference of a single moiré exciton in nano-fabricated twisted monolayer semiconductor heterobilayersREVIEWER COMMENTS

Reviewer #1 (Remarks to the Author):

The authors in their paper titled "Quantum coherence and interference of a single moiré exciton in nano-fabricated twisted semiconductor heterobilayers" demonstrate quantum coherence and interference in single moire excitons in vdW heterostructures of MoSe₂ and WSe₂. Moire excitons in heterostructure bilayer systems have been observed before. However, as rightly pointed out by the authors "experimentally, the important information on the quantum coherence and interference of moiré excitons in the vdW heterobilayers remains unexplored, because the overlapping of multiple emission peaks from moiré excitons in the inhomogeneously broadened spectra hinders these intrinsic insights within the diffraction limit of light." This work potentially addresses the issue by lithographically patterning nanopillars thus making it possible to isolate emissions and reduce inhomogeneous broadening effects from multiple emission peak from Moire excitons. This strategy of fabricating nanopillars has previously been demonstrated by the authors in their previous paper "Dynamics of Moiré Trion and Its Valley Polarization in a Microfabricated WSe₂/MoSe₂ Heterobilayer" but the study of quantum coherence and interference of moiré exciton in this context was unexplored and the authors shed light on that aspect in this paper. The work overall seems quite interesting, and I recommend publishing it in Nature Communications provided the authors address some of the questions discussed below.

1. In the previous work referenced above, the authors used a similar strategy to isolate/localize excitation and emission by fabricating nanopillars. However, in this paper RIE with Ar gas is used instead of the Ga⁺ ion beams used in the previous paper. Can the authors point out why this change was made? Was it just logistics or is there a scientific reason for this?
2. Power dependent studies in figure 2, seem to be done at specific discrete values of power i.e. 0.8 W/cm², 1.6 W/cm², 3.1 W/cm²... Is this accurate? From the contour plot, it seems like there is data for every power density in between. Can the authors clarify and make the necessary changes to address this?
3. In Fig 2e, two fittings corresponding to the two peaks (moire exciton and biexciton) the power dependence are shown. The y- axis on both sides seems to be in Logarithmic scale. Would it make more sense to change it to a linear scale on one side. That way the authors can show the linear power dependence clearly for the moire exciton on a linear scale and non-linear power dependence for the biexciton on a logarithmic scale.
4. Also, since the linear dependence starts showing saturation at higher powers, it might be a good idea to fit the linear points (first 3), where there is no saturation instead of fitting all the points with a linear plot.
5. Once would expect the authors would see similar effects with the reverse bilayer (WSe₂ on top instead of MoSe₂ and hence reversing the order)? If so, is it worthwhile to do the same experiments as proof of concept?

Reviewer #2 (Remarks to the Author):

Report on manuscript entitled "Quantum coherence and interference of a single moiré exciton in nano-fabricated twisted semiconductor heterobilayers"

In this manuscript, Haonan Wang and coauthors claim they have developed a new microfabrication method to realize the optical observation of quantum coherence and interference of a single moiré exciton in twisted semiconducting heterobilayer (HB) beyond the diffraction limit of light. Their approach is based on the use of electron beam lithography and selective reactive ion etching to etch the HB samples in the shape of pillars. After microfabrication, the authors conducted photoluminescence experiments to show a single peaked interlayer exciton emission from their samples and claim, without further discussion, that this emission comes from moiré excitons. In a further step, the authors observed the quantum coherence of such interlayer excitons by the use of a Michelson interferometer and discuss the factors that influence that coherence (intrinsic exciton lifetime and decoherence mechanisms). Moreover, in a sample in which the authors observe a double peak structure, they conduct experiments to show quantum beating and

coupling between those interlayer excitons.

The data are sound and the analysis is, as far as can be judged, solid. However, to the best of my knowledge, the observation of the interlayer exciton emission does not imply the presence of a moiré pattern. On the other hand, the study presents a rather incremental advance of specific knowledge about interlayer excitons but lacks the criteria of impact, novelty and generality that would make it suitable for publication in Nature Communications. I recommend transferring it to a more specialized journal.

Below are my detailed comments that need to be corrected or further discussed in the manuscript:

1. The isolation and manipulation of interlayer excitons with similar structures to those presented in this work have been performed and reported in earlier papers like, for instance, [npj 2D Materials and Applications 4, 8 (2020), Communications Physics 4, 119 (2021)] but are not mentioned in the text.
2. The authors claim that the emission they observe stems from moiré excitons, i.e. interlayer excitons confined in moiré traps. While this discussion does not affect their observations and conclusions, this strong statement is not correct or at least sufficiently discussed in the manuscript. The moiré pattern is extremely sensitive to the contact between layers and other extrinsic conditions like, e.g. strain [Nature Materials 19, 1068 (2020), npj 2D Materials and Applications 7, 32 (2023)]. Moreover, when the stacking angle is close to 0° or 60° , minor external manipulation, like thermal cycling or strain during fabrication produces reconstruction [A. Weston et al, Nat. Nanotech. 15, 592 (2020); T. I. Andersen et al, Nat. materials 20, 480 (2021); V. Enaldiev et al, PRL 124, 206101 (2020)]. This effect implies the local deformation or the local rotation of the TMD monolayers, within the moiré sites, that results in the relaxation of the moiré potential. In my opinion, it is not clear if the use of ion etching during the pillar fabrication stresses or affects the TMD monolayers. Moreover, it has been shown that interlayer excitons present similar emission even in the total absence of moiré patterns [Nat Comm. 13, 5354 (2022)].
3. The stacking angle of the HB is not properly discussed. In TMD materials, the lack of inversion symmetry makes the identification of the stacking angle undetermined, i.e., 0° and 60° looks the same but are completely different from the band alignment and photoemission. The correct way to determine the stacking angle is done, for instance, via SHG experiments. In the case of WSe₂/MoSe₂ HBs, 0° or 60° samples display different optical selection rules. Therefore, polarized resolved PL can be used to determine if the sample is close to 0° or 60° . Moreover, the selection rules observed in these HBs can be used to check the presence of moiré patterns [ACS Nano 16, 16862 (2022), Phys. Rev. X 11, 031033 (2021), Nat Comm. 13, 5354 (2022)]. Such a measurements could help the authors to answer the previous point.
4. The authors should carefully improve the English grammar. For example, there is a recurrent lack or excess of articles (a, the), and the use of the adjective form instead of the adverb form and vice versa (e.g., optical instead of optically and vice versa).
5. Finally, last paragraph of page 10 cites eq.(3), but I think it should be eq.(2).

Reviewer #3 (Remarks to the Author):

In "Quantum coherence and interference of a single moiré exciton in nano-fabricated twisted semiconductor heterobilayers", the authors present optical spectroscopy and quantum optics measurements of single, spatially localized interlayer excitons in WSe₂/MoSe₂ heterobilayers deposited on top of nanopillars with diameters in the range 50 – 500 nm. For the narrowest pillar used (50 nm), photoluminescence measurements at a temperature of 4 K reveal the presence of a single dominant PL emission line with features consistent with 0-D confinement: i) the typical

saturation behavior of a two-level system; and ii) spectral wandering due to charge noise in the vicinity of the spatially localized exciton. The power-dependent evolution of the PL spectrum (which reveals the presence of biexcitons at higher excitation powers) allows to attribute the relatively narrow PL emission line observed at low exciton densities to the emission of a single localized exciton. The authors further perform first-order correlation function measurements of the narrow PL emission line using a Michelson interferometer and estimate the coherence time as a function of temperature and exciton density. Combined with the results from time-resolved PL measurements, the authors estimate the pure dephasing time of the localized emitter. Finally, the authors perform similar first-order correlation function measurements in a second pillar, in which two PL emission lines with similar intensities and separated by ~ 18 meV are observed at the lowest excitation power shown by the authors. The first-order correlation function measurements reveal an interferogram beating with a period of 230 ps, which corresponds to an energy splitting that agrees well with the observed energy difference between the emission lines in the PL spectra. The authors attribute such quantum beating to the coupling of the two localized interlayer excitons, which they claim originate from two moiré sites with different potential depths. Overall, I find the study presented by the authors quite relevant for the community of researchers working in TMD heterostructures. Despite the several previous demonstrations of single-moiré-trapped interlayer exciton emission in WSe₂/MoSe₂ heterobilayers and heterotrilayers by different groups, to the best of my knowledge the quantum coherence and the pure dephasing time of these photon emitters has not previously been reported. Based on all this, I would like to recommend this work for publication in Nature Communications. However, I have a series of comments/questions that I believe the authors should address before my final recommendation for publication.

Below are my detailed comments:

1. I would encourage the authors to elaborate a bit more on their claim that they “have demonstrated a new method to realize the optical observation of quantum coherence and interference of a single (...) exciton (...) beyond the diffraction limit of light.” What is it exactly that makes the method novel (if it is indeed novel)? To my knowledge, the use of nanopillars has already been used in the past to isolate single spatially localized excitons in both individual 2D materials (see Nature Commun. 8, 15093 (2017) and Nature Commun. 8, 15053 (2017) among many others) and MoSe₂/WSe₂ heterostructures (npj 2D Mater. Appl. 4, 8 (2020)). In a similar way, carrying first-order correlation function measurements of the emitted photons using a Michelson interferometer is also a standard way to characterize the coherence time. Therefore, I think it would be good for the manuscript if the authors could clarify/expand why they think their approach is novel.
2. Following on the use of nanostructure fabrication to achieve emission from a single localized interlayer exciton, do the authors have any statistics regarding the frequency with which they obtain a single localized interlayer exciton per confocal optical spot? It seems to me that the PL spectra measured at the pillar used in the first part of the work also shows PL from at least 2 different excitons (~ 1.38 and ~ 1.39 eV) even at the lowest excitation shown in the paper (although with the exciton at ~ 1.38 eV being considerably brighter than the other one).
3. Throughout the work, the authors claim the moiré superlattice to be responsible for the spatial confinement of their interlayer excitons. What evidence do the authors have for this? I note that, contrary to previous works on moiré-trapped interlayer excitons in MoSe₂/WSe₂, the authors do not present any evidence about the selection rules nor the magnetic properties characteristic of moiré-trapped interlayer excitons (see Nature 567, 66 (2019)). Moreover, for the twist angle estimated from the optical images ($\sim 1^\circ$), one should expect a strong lattice reconstruction for both 2H- and 3R-stacked MoSe₂/WSe₂ heterobilayers. Do the authors know their relative stacking orientation (i.e., 2H vs 3R)? Also, it's worth noting that for nearly lattice-matched 2D heterobilayers such as MoSe₂/WSe₂ there are theoretical predictions of exciton localization in the hot spots between a network of domain walls separating domains of commensurate stacking (see npj 2D Mater. Appl. 6, 74 (2022)). For all these reasons, I think it would be more appropriate to refer to the excitons in this work as “spatially localized interlayer excitons” rather than “moiré excitons”, since the authors do not prove their moiré origin.

4. In the interferograms measured for different temperatures (or exciton densities), the periodicity of the interference fringes does not seem to evolve monotonically with the temperature (or excitation power). For example, in Suppl. Fig 9, the periodicity of the fringes seems to be longer for 80 W/cm² than for 14 W/cm², but it seems to decrease again at 200 W/cm². A similar behavior is observed in Suppl. Fig. 10. Could the authors explain the origin of this?

5. How do the authors estimate the visibility of the interferences in the interferograms as a function of delay time? The amplitude of the maxima and minima also changes with the delay time. Do the authors use consecutive maxima and minima? Or do they average within a given time window? I think the authors could add a sentence explaining how this is done.

6. In the TRPL results, the authors explain that the measured curves follow a tri-exponential decay, from which they extract the three relevant decay times. These decay times go from 14 ns up to 700 ns at 4K. Then, the authors estimate an "average lifetime for the moiré exciton". Can the authors explain how they estimate such an average time and what it's its physical meaning?

7. On a related comment, I think the authors should comment on which lifetime T1 is being used in equation (3) to estimate the pure dephasing rate (and why).

8. Finally, I think the authors could have done a better job referencing relevant previous work. It is surprising that, despite the authors' claim of the moiré lattice being responsible for the interlayer exciton trapping, they do not cite the pioneering work from Xiadong Xu's lab on the first observation of moiré-trapped interlayer excitons (Nature 567, 66 (2019)). In a similar way, it is also surprising the omission of references to other previous works on moiré excitons, such as the demonstration of their single-photon emitter (quantum) nature (Sci. Adv. 6, eaba8526 (2020)), or their evolution from the single-quantum emitter to the interlayer exciton ensemble regime (Phys. Rev. X 11 (3), 031033 (2021)), among many others.

Finally, below are some additional comments:

1. I believe the order (labelling) of supplementary figures S9 and S10 is reversed with respect to the description in the main text.

2. The authors state that the narrowest emission linewidth (~600 micro eV) is almost limited by the spectral resolution of the system. However, from panel Fig. 2c I can see that the narrowest emission line contains at least ~9 data points, which suggests to me that the PL lines are far from being resolution limited. This can also be seen in the noise of the PL background, where it's obvious that the resolution is better than 600 micro eV.

RESPONSE TO REVIEWERS' COMMENTS

Figure updates:

1. We have evaluated the error bars of the data and added to Fig. 1(d), Fig. 3(h), and 3(j).
2. Figure 2(e) has been updated with new linear fitting according to Reviewer's advice.
3. We have conducted additional SHG experiments in monolayer MoSe₂, WSe₂ and heterobilayer region, and been added the new data in Supplementary Fig. 1(b).
4. The plots of peak intensities as function of excitation power with linear scale and logarithmic scale have been added as Supplementary Fig. 4.
5. The PL spectra from pillars in different positions have been added as Supplementary Fig. 6.
6. We have added the new data of optical image, PL spectra and SHG of new heterobilayer sample, with reversed stacking in Supplementary Fig. 7.
7. The experimental results of circular polarization-resolved PL spectra as a function of excitation power have been added in Supplementary Fig. 8.
8. We also added additional experiment results of linearly polarized PL spectra as a function of excitation power in Supplementary Fig. 9.
9. The description of pillar size in the caption of Fig. 2(b) has been revised.

Summary of changes:

10. The descriptions regarding previous works of isolation and manipulation of interlayer exciton have been added in the introduction part of main text in Page 5. The related reference papers have also been added according to this revision.
11. The caption of Fig. 2(a) with clarification of the contour plot has been changed.
12. The description about the twist angle of the heterobilayer has been revised in the last paragraph in Page 9. The new explanation based on the SHG experiment has been added.
13. The supplementary Note 2 has been revised for estimation of exciton density.
14. We have added the description about linear and circular polarization-resolved PL spectra of heterobilayer in Page 10 as the experimental evidences of moiré-exciton emission.
15. The experiment procedure of SHG measurement has been added in the Supplementary Note 1.
16. We have added the explanation for determining the maximum and minimum intensity from the interferograms in Line5, Page 12.
17. The citation of equation (3) has been revised in Page 12.

18. The explanation of decay lifetime used for estimating pure dephasing time has been added in the first paragraph of Page 13.
19. The citations of Supplementary Fig. 14 and Supplementary Fig. 15 have been exchanged in Page 14 and Page 15.
20. We have revised the description of spectral resolution.
21. We have added the references according to Reviewer's suggestions.
22. We have added monolayer in the title.

Point by point response to reviewers:

We are thankful for your helpful comments and elaborated suggestions. We have considered the Reviewers' comments carefully to further improve our manuscript for the publication in *Nature Communications*.

The revisions have been made to the manuscript following the Reviewers' comments. We hope that we have now produced a better account for our work. We would like to resubmit our revised manuscript to *Nature Communications* for publication. We studied the Reviewers' comments carefully and the revisions made are described as follows. We introduced additional data following the advices received from the Reviewers.

Reply to the comments of Reviewer #1

The authors in their paper titled “Quantum coherence and interference of a single moiré exciton in nano-fabricated twisted semiconductor heterobilayers” demonstrate quantum coherence and interference in single moiré excitons in vdW heterostructures of MoSe₂ and WSe₂. Moiré excitons in heterostructure bilayer systems have been observed before. However, as rightly pointed out by the authors “experimentally, the important information on the quantum coherence and interference of moiré excitons in the vdW heterobilayers remains unexplored, because the overlapping of multiple emission peaks from moiré excitons in the inhomogeneously broadened spectra hinders these intrinsic insights within the diffraction limit of light.” This work potentially addresses the issue by lithographically patterning nanopillars thus making it possible to isolate emissions and reduce inhomogeneous broadening effects from multiple emission peak from moiré excitons. This strategy of fabricating nanopillars has previously been demonstrated by the authors in their previous paper “Dynamics of Moiré Trion and Its Valley Polarization in a Microfabricated WSe₂/MoSe₂ Heterobilayer” but the study of quantum coherence and interference of moiré exciton in this context was unexplored and the authors shed light on that aspect in this paper. The work overall seems quite interesting, and I recommend publishing it in Nature Communications provided the authors address some of the questions discussed below.

We would like to thank the Reviewer for careful reading and summary of our manuscript. We have cautiously thought about the Reviewer's questions. Our answers and revisions are described below.

1. In the previous work referenced above, the authors used a similar strategy to isolate/localize excitation and emission by fabricating nanopillars. However, in this paper RIE with Ar gas is used instead of the Ga⁺ ion beams used in the previous paper. Can the authors point out why this change was made? Was it just logistics or is there a scientific reason for this?

As described by the Reviewer, the FIB method has been used in the previous work where we apply the relatively high energy Ga⁺ ion beam to the heterobilayer structure

for the fabrication of nano-pillars. A series of pillars with different sizes have been made, as shown in the previous paper. However, in the measurement of FIB-treated heterostructures, it is found that the PL intensity quickly decreases with the size under 200 nm due to the edge effect caused by diffused Ga^+ ion. The nano-pillar size of 200 nm is the smallest value that can be achieved to avoid the reduction of optical properties in the heterostructures. The relative high energy Ga^+ ion beam makes the damage layers in the nano-pillar with the wall region of about 30 nm during the process, which has been confirmed by observation of transmission electron microscope (TEM) image, as shown in Fig. R1.

In contrast, in the RIE method, it is found that the damage by reactive ions in Ar gas to the treated area is almost negligible, leading to considerable PL intensity even under much smaller diameter of 50 nm, which is now mainly limited by the resolution of our electron-beam lithography. Thus, this advanced new method expands the size limitation of nanopillar, which also makes to realize the observation of single moiré exciton signals in this study.

Figure R1 Typical TEM image of FIB treated hetero-structures. (a) TEM image of FIB treated region of a heterostructure. **(b)** Enlarged image of the region indicated by green square in (a). **(c)** Enlarged TEM image of the crystal region in (b). **(d)** Enlarged TEM image of the amorphous region caused by FIB.

2. Power dependent studies in figure 2, seem to be done at specific discrete values of power i.e. 0.8 W/cm², 1.6 W/cm², 3.1 W/cm².... Is this accurate? From the contour plot, it seems like there is data for every power density in between. Can the authors clarify and make the necessary changes to address this?

We measured the power-dependent PL spectrum under discrete power densities, as pointed out by the Reviewer. Figure 2(a) shows the continuous counter plot using the

linear interpolation method, however we confirmed that the data of PL spectra have been measured under enough small intervals of power density. We use such contour plot for the purpose of better demonstration of the intensity-saturation behaviors and blue-shift of the spectra. The similar continues contour plot have been used in the previous paper [*Phys. Rev. X* **11** (3), 031033 (2021), *npj 2D Mater. Appl.* **5**, 67 (2021), and *npj 2D Mater. Appl.* **4**, 8 (2020)].

We added the description of detail of data plot to avoid the misunderstanding of readers (update #11).

3. In Fig 2e, two fittings corresponding to the two peaks (moiré exciton and biexciton) the power dependence are shown. The y- axis on both sides seems to be in Logarithmic scale. Would it make more sense to change it to a linear scale on one side. That way the authors can show the linear power dependence clearly for the moiré exciton on a linear scale and non-linear power dependence for the biexciton on a logarithmic scale.

According to the Reviewer suggestion, we also added the plot of PL intensity as a function of excitation power density of the two-peaks (moiré exciton and biexciton) with logarithmic and linear scale, as shown in Fig. R2 (also see Supplementary Fig. 4).

Figure R2 PL intensities of peaks at 1.380 and 1.382 eV as a function of excitation power density with logarithmic and linear scale, with x-axis being the logarithmic scale (a) and linear scale (b).

4. Also, since the linear dependence starts showing saturation at higher powers, it might be a good idea to fit the linear points (first 3), where there is no saturation instead of fitting all the points with a linear plot.

Figure R3 PL intensities of peaks at 1.380 and 1.382 eV as a function of excitation power density. The black line indicating linear dependence is refitted.

We appreciate the suggestion from the Reviewer. Several data points with signature of saturation behavior above the saturation power density have been used for the fitting of data plot. We have refitted the plots in Fig. 2(e) with only data points below the saturation power density, and revised the Fig. 2(e) (update #2).

5. Once would expect the authors would see similar effects with the reverse bilayer (WSe₂ on top instead of MoSe₂ and hence reversing the order)? If so, is it worthwhile to do the same experiments as proof of concept?

We have additionally made the another heterobilayer sample with reversed stacking order of monolayer WSe₂ and MoSe₂ (*h*-BN/ WSe₂/MoSe₂/*h*-BN), as shown in Fig. R4(a). The experimental results of SHG in Fig. R4(d) indicate that the heterobilayer shows the twist angle of 55.8 degree with H-type stacking. Moreover, the same RIE procedure has been applied to this heterobilayer sample.

We have also conducted the additional PL measurements under the same excitation laser and low temperature conditions. We can clearly see the sharp and isolated PL peaks from moiré excitons at 1.329 eV under the lower power conditions below 31 W/cm² in Fig. 4(b), which are similar experimental results of in Fig. 2(b) in the main text. Moreover, the appearance of biexciton peak with showing the nonlinear behavior as a function of excitation power at 1.331 eV have also been clearly observed in Fig. R4(b). Thus, we can confirm the same experimental results in the heterobilayer (*h*-BN/ MoSe₂/WSe₂/*h*-BN) in the main text, which indicates that the experimental results studied here show the common and universal results in the heterobilayer independent on the stacking order of monolayers.

Figure R4 Heterobilayer sample with reversed stacking order of WSe₂ and MoSe₂. (a) Optical image of the WSe₂/MoSe₂ heterobilayer after the same RIE process. (b) Low temperature PL spectra normalized by the excitation powers under various excitation powers. (c) Integrated peak intensities as a function of excitation power. (d) Polar plot of SHG intensities of the WSe₂/MoSe₂ heterobilayer (red), monolayer WSe₂ (green), and MoSe₂ (black) region.

Reply to the comments of Reviewer #2

In this manuscript, Haonan Wang and coauthors claim they have developed a new microfabrication method to realize the optical observation of quantum coherence and interference of a single moiré exciton in twisted semiconducting heterobilayer (HB) beyond the diffraction limit of light. Their approach is based on the use of electron beam lithography and selective reactive ion etching to etch the HB samples in the shape of pillars. After microfabrication, the authors conducted photoluminescence experiments to show a single peaked interlayer exciton emission from their samples and claim, without further discussion, that this emission comes from moiré excitons. In a further step, the authors observed the quantum coherence of such interlayer excitons by the use of a Michelson interferometer and discuss the factors that influence that coherence (intrinsic exciton lifetime and decoherence mechanisms). Moreover, in a sample in which the authors observe a double peak structure, they conduct experiments to show quantum beating and coupling between those interlayer excitons.

The data are sound and the analysis is, as far as can be judged, solid. However, to the best of my knowledge, the observation of the interlayer exciton emission does not imply the presence of a moiré pattern.

We would like to appreciate the Reviewer's careful reading of our manuscript. Regarding the origin of the observed emission signal, we have conducted additional many experiments and added the discussion for identify that it comes from moiré excitons.

On the other hand, the study presents a rather incremental advance of specific knowledge about interlayer excitons but lacks the criteria of impact, novelty and generality that would make it suitable for publication in Nature Communications. I recommend transferring it to a more specialized journal.

We would like to emphasize the impact, novelty and generality of our findings.

The interlayer moiré exciton system with novel optical properties is expected to be a promising platform for the development of quantum technology. However, due to the inhomogeneity among periodic moiré potentials, the experimentally observed multiple emission peaks from moiré excitons within the diffraction limit of light conceals the most important and essential information of quantum coherence in the moiré exciton, which have hindered the application of quantum optics based on moiré exciton system.

In this study, we present a novel nanofabrication strategy to realize the observation of a single moiré exciton beyond the diffraction limit of light, enabling direct measurement of coherence time of this new quantum two level-system and exploration of its dephasing mechanism. Furthermore, we successfully observe the coherent coupling between different moiré excitons revealed by quantum beat signals in the temporal interferogram. Our research highlights the unique characteristics of coupled quantum systems based on moiré excitons within semiconducting heterobilayer, which can be extended to manipulate multiple quantum states within periodic moiré superlattice.

Our investigation into the quantum coherence and interference of moiré excitons lays

the groundwork for applying the moiré system to quantum optical technology. The new findings regarding long coherence time and quantum coupling could bring new insights to quantum computing, quantum communication, and quantum cryptography for the future. We believe that our findings in this study are well satisfied higher standard in terms of scientific impact, novelty and generality to publish in Nature Communications.

1. The isolation and manipulation of interlayer excitons with similar structures to those presented in this work have been performed and reported in earlier papers like, for instance, [npj 2D Materials and Applications 4, 8 (2020), Communications Physics 4, 119 (2021)] but are not mentioned in the text.

As pointed out by the Reviewer, some important previous works are not described in the main text and cited in the reference. According to the Reviewer advice, we have made the revisions in the main text and reference parts (update #10).

2. The authors claim that the emission they observe stems from moiré excitons, i.e. interlayer excitons confined in moiré traps. While this discussion does not affect their observations and conclusions, this strong statement is not correct or at least sufficiently discussed in the manuscript. The moiré pattern is extremely sensitive to the contact between layers and other extrinsic conditions like, e.g. strain [Nature Materials 19, 1068 (2020), npj 2D Materials and Applications 7, 32 (2023)]. Moreover, when the stacking angle is close to 0° or 60° , minor external manipulation, like thermal cycling or strain during fabrication produces reconstruction [A. Weston et al, Nat. Nanotech. 15, 592 (2020); T. I. Andersen et al, Nat. materials 20, 480 (2021); V Enaldiev et al, PRL 124, 206101 (2020)]. This effect implies the local deformation or the local rotation of the TMD monolayers, within the moiré sites, that results in the relaxation of the moiré potential. In my opinion, it is not clear if the use of ion etching during the pillar fabrication stresses or affects the TMD monolayers. Moreover, it has been shown that interlayer excitons present similar emission even in the total absence of moiré patterns [Nat Comm. 13, 5354 (2022)].

Thank you for your helpful and important comments. As the Reviewer pointed out, the description, and discussion about the emission arising from moiré excitons, i.e. interlayer excitons confined in moiré traps, does not strongly affect their observations and conclusions. However, we have conducted additional experiments such as polarization dependent and its power dependence of PL spectra for further clarifying the origin of the signals, as following the previous studies [*Nature* 567, 66 (2019), and *Phys. Rev. X* 11, 031033 (2021)].

We measured polarization dependence of PL spectra from a single nano-pillar in the heterobilayer. Figure R5(a) shows the PL spectra excited by linearly polarized pulsed laser source with repetition rate of 5 MHz and excitation energy of 1.72 eV at 4 K. The PL spectra do not show large changes with varying the linear-polarization angle. Moreover, the polar plot of PL intensity in Fig. R5(b) shows the polarization independent behavior. These results suggest that the emission comes from the unstrained moiré potential with C_{3v} symmetry in the heterobilayer with nano-fabrication

process, because the previous studies have demonstrated the strong linear-polarization behavior of exciton emissions in the strain-induced distorted more potentials.

We also measured the circular polarization dependent PL spectra under the excitation photon energy of energy of 1.72 eV at 4 K, where the excitation photon energy corresponds to the exciton resonance in monolayer WSe₂. Figure R6(a) shows PL spectra from the nano-pillar with a size of 50 nm under co- and cross-circular polarization excitation and detection configurations. The PL spectrum shows the stronger emission signals under co-polarized condition than cross-polarized condition. These characteristics circularly-polarization dependent PL spectrum reflected by the optical selection rule also indicates that emission come from exciton-trapped in the moiré potentials in the heterobilayer with H-type stacking.

Figures R6(a) and (b) show polarization resolved PL spectra with varying the excitation power. The PL peak intensity difference between co- and cross-circular polarization configuration changes strongly depending on the excitation power. The degree of circularly polarization (DOCP) defined as $(I_{co} - I_{cross}) / (I_{co} + I_{cross})$ is plotted

Figure R5 (a) Contour plot of PL spectrum with changing the linear polarization angle of excitation laser. (b) PL intensity of the emission signal as a function of the linear detection angle.

in Fig. R6(b), and shows the excitation power dependent behaviors, which is consistent with the previous result of emission from moiré exciton trapped in the moiré potential in the heterobilayer without nano-fabrication process [*ACS Nano* **16**, 16862 (2022)]. The strong power dependent DOCP also suggests that the emission comes from moiré exciton trapped in the potential with smaller 0D-like density of states, rather than interlayer exciton with larger 2D-like density of states.

Finally, we will also discuss the physical origin of emission (0D-like moiré exciton or 2D-like interlayer exciton) from the exciton coherence. The previous studies on the

Figure R6 (a) PL spectrum of co- and cross-polarization configuration under circularly polarized excitation and detection with different excitation power at 4 K. (b) Degree of circular polarization (DOCP) as a function of excitation power.

2D exciton from the monolayer transition metal dichalcogenides and quantum well systems have demonstrated that the excitonic coherence time is limited below 10 ps (130 μeV) even under zero-temperature. [*Phys. Rev. Lett.* **67**, 2355 (1991), *Science* **273**, 87 (1996)]. In contrast, the experimentally obtained longer exciton coherence of 12 ps ($<100 \mu\text{eV}$) in this study shows the overcomes of value in the typical 2D-systems, which strongly suggests that emission signals come from the moiré exciton trapped in the potential, rather than interlayer exciton. Moreover, the moiré excitons in the heterobilayer show the intrinsic properties, not affected by the local deformation and/or the local rotation that results in the relaxation of the moiré potential during the nano-fabrication process.

Furthermore, we have conducted SHG measurement for precisely determination of the twist angle and stacking type, as also suggested by the Reviewer in Q3. The SHG results in Fig. R7 indicates the twist angle of 56.5 ± 0.3 degree with H-type stacking, which is well consistent with the characteristic results of polarization resolved PL spectroscopy that co-polarized intensity shows stronger intensity than cross-polarized spectra, as shown in Fig. R6(a). As the Reviewer pointed out, the effect of lattice reconstruction in twisted $\text{MoSe}_2/\text{WSe}_2$ heterobilayer with near 0 and 60 degrees (twist angle < 3 degree in R-type and H-type) will cause the exciton localization and 0D-featured optical properties. However, we can confirm that the twist angle in the heterobilayer in this study is > 3 degree, which is not in the case.

Finally, we would like to describe about the suggested paper [*Nat. Commun.* **13**, 5354 (2022)], in which assemblies of sharp emission lines are observed even when the moiré

pattern is suppressed by insertion of thin *h*-BN (inter-layer exciton). It should be noted that the circular-polarized emission from inter-layer exciton shows opposite-helicity compared with that of the moiré exciton, and significant blue-shifted spectrum center (1.42 eV) in comparison with the moiré exciton (1.34 eV) in R-type stack, which is much contradict to our experimental results, as shown in Figure 1(c) and Figure 2(b).

Thus, we can confirm that the single emission peak observed in our experiment is from the exciton trapped in moiré potential based on above experiment results and discussion.

3. The stacking angle of the HB is not properly discussed. In TMD materials, the lack of inversion symmetry makes the identification of the stacking angle undetermined, i.e., 0° and 60° looks the same but are completely different from the band alignment and photoemission. The correct way to determine the stacking angle is done, for instance, via SHG experiments. In the case of WSe₂/MoSe₂ HBs, 0° or 60° samples display different optical selection rules. Therefore, polarized resolved PL can be used to determine if the sample is close to 0° or 60° . Moreover, the selection rules observed in these HBs can be used to check the presence of moiré patterns [ACS Nano 16, 16862 (2022), Phys. Rev. X 11, 031033 (2021), Nat Comm. 13, 5354 (2022)]. Such a measurements could help the authors to answer the previous point.

Figure R7 Polarization-resolved SHG measurement. The six-fold symmetry in the SHG intensity (points) and fits (solid lines) for the MoSe₂ monolayer (black), WSe₂ monolayer (green) and MoSe₂/WSe₂ heterobilayer (red). SHG result of WSe₂ monolayer shows relatively larger experimental errors, which is caused by limited monolayer area after the nanofabrication. SHG measurements are conducted in the unetched areas.

We would like to thank the Reviewer for the helpful suggestions. We also think that the identification of stacking angle of the heterobilayer should be properly measured by SHG experiments. According to the Reviewer advice, the experiments of SHG on our heterobilayer sample have been conducted. The polar patten of SHG signals and their significantly weakened intensity in the heterobilayer in comparison with constitutive

monolayers in Fig. R7 shows the H-type stacking which corresponds to the circularly-polarized PL spectroscopy reflected by the optical selection rules (Fig. R6). The fitted results of the six-fold polar patterns indicate the relative twist angle of the heterobilayer is measured to be $56.5\pm 0.3^\circ$. Based on the precise determination of twist angle results by additional SHG experiments, we have carefully checked the estimation process and made revision in the Supplementary Note 2 and in the main text (update #3, #12, and #13).

4. The authors should carefully improve the English grammar. For example, there is a recurrent lack or excess of articles (a, the), and the use of the adjective form instead of the adverb form and vice versa (e.g., optical instead of optically and vice versa).

Thank you for careful reading of our manuscript and suggesting the appropriate revision. The manuscript has been sent to professional English correction service for checking, and revised.

5. Finally, last paragraph of page 10 cites eq. (3), but I think it should be eq. (2).

As pointed out by the Reviewer, the correction has been made in the revised manuscript (update #17).

Reply to the comments of Reviewer #3

In “Quantum coherence and interference of a single moiré exciton in nano-fabricated twisted semiconductor heterobilayers”, the authors present optical spectroscopy and quantum optics measurements of single, spatially localized interlayer excitons in WSe₂/MoSe₂ heterobilayers deposited on top of nanopillars with diameters in the range 50 – 500 nm. For the narrowest pillar used (50 nm), photoluminescence measurements at a temperature of 4 K reveal the presence of a single dominant PL emission line with features consistent with 0-D confinement: i) the typical saturation behavior of a two-level system; and ii) spectral wandering due to charge noise in the vicinity of the spatially localized exciton. The power-dependent evolution of the PL spectrum (which reveals the presence of biexcitons at higher excitation powers) allows to attribute the relatively narrow PL emission line observed at low exciton densities to the emission of a single localized exciton. The authors further perform first-order correlation function measurements of the narrow PL emission line using a Michelson interferometer and estimate the coherence time as a function of temperature and exciton density. Combined with the results from time-resolved PL measurements, the authors estimate the pure dephasing time of the localized emitter. Finally, the authors perform similar first-order correlation function measurements in a second pillar, in which two PL emission lines with similar intensities and separated by ~18 meV are observed at the lowest excitation power shown by the authors. The first-order correlation function measurements reveal an interferogram beating with a period of 230 ps, which corresponds to an energy splitting that agrees well with the observed energy difference between the emission lines in the PL spectra. The authors attribute such quantum beating to the coupling of the two localized interlayer excitons, which they claim originate from two moiré sites with different potential depths.

We thank the Reviewer for careful reading and comprehensive summary of our manuscript.

Overall, I find the study presented by the authors quite relevant for the community of researchers working in TMD heterostructures. Despite the several previous demonstrations of single-moiré trapped interlayer exciton emission in WSe₂/MoSe₂ heterobilayers and heterotrilayers by different groups, to the best of my knowledge the quantum coherence and the pure dephasing time of these photon emitters has not previously been reported. Based on all this, I would like to recommend this work for publication in Nature Communications. However, I have a series of comments/questions that I believe the authors should address before my final recommendation for publication.

We would like to thank the Reviewer for significantly positive evaluation of our study. We have cautiously thought about the Reviewer’s question. Our answers and revisions are described below.

1. I would encourage the authors to elaborate a bit more on their claim that they “have demonstrated a new method to realize the optical observation of quantum

coherence and interference of a single (...) exciton (...) beyond the diffraction limit of light.” What is it exactly that makes the method novel (if it is indeed novel)? To my knowledge, the use of nanopillars has already been used in the past to isolate single spatially localized excitons in both individual 2D materials (see *Nature Commun.* **8**, 15093 (2017) and *Nature Commun.* **8**, 15053 (2017) among many others) and MoSe₂/WSe₂ heterostructures (*npj 2D Mater. Appl.* **4**, 8 (2020)).

In a similar way, carrying first-order correlation function measurements of the emitted photons using a Michelson interferometer is also a standard way to characterize the coherence time. Therefore, I think it would be good for the manuscript if the authors could clarify/expand why they think their approach is novel.

Thank you for your important and helpful comments. We would like to carefully explain the significantly difference between our approach and previously reported ones, regarding the novelty of our nano-fabrication method, and first-order correlation function measurement.

As the Reviewer pointed out, some papers have reported introducing localized-strain by micro(nano)-pillars to generate spatially localized excitons in semiconducting monolayer two-dimensional (2D) materials. These methods are effective, yet, introducing the strain-induced localized exciton and incoherent energy transfer process between delocalized and localized excitons. Such processes would conceal some intrinsic properties, such as the valley degree of freedom, dipole moment, dephasing time and so on. Moreover, most of the studies are concentrated on the excitons in monolayer 2D materials, which is much different from study of moiré exciton in the heterobilayer.

In contrast, we would like to emphasize that our method maintains the intrinsic optical properties of moiré exciton even after the process in the semiconducting heterobilayer. We can prove that the RIE method introduces much less edge effect to the etched area, by demonstrating PL spectrum from the nano-pillar size with ranging from 50 to 500 nm. Moreover, this novel approach allows utilizing the advantageous properties of moiré exciton, such as long lifetime (Supplementary Fig. 13), helicity-selective optical response (Fig. R6) and longer intrinsic coherence time (Fig. 3 and Fig. 4).

As the Reviewer pointed out, the first-order correlation measurement using Michelson interferometer has been a standard method for the characterization of coherence properties, However, such a direct measurement can be limited in certain isolated quantum systems, mainly atomic (molecular) system, or single semiconductor quantum dots [*Science* **282**,1473 (1998), *Phys. Rev. Lett.* **87**, 246401 (2001)]. The directly applying first-order correlation measurement can be difficult to precisely determine the coherence time for moiré exciton systems, due to the large inhomogeneous broadening of the ensembled-emissions in the heterobilayer.

Actually, the characterizations of coherence feature in monolayer 2D semiconductors have been previously conducted by four-wave-mixing (FWM) or 2D-coherent spectroscopy with strong excitation power conditions reaching to non-linear optical regime [*Phys. Rev. Lett.* **116**, 127402 (2016), *Phys. Rev. Mater.* **2**, 054001 (2018), *Nat.*

Commun. **7**, 13279 (2016)]. However, the above-mentioned methods such as FWM are not technically applicable due to the relatively low oscillator strength of moiré-trapped interlayer exciton. Moreover, the first-order correlation function should be measured to determine the coherence time by an emission signal from a single moiré exciton, not ensemble-averaged signals from many moiré excitons, in which we have demonstrated by novel approach in this study. Thus, at the present stage, the first-order correlation measurement using Michelson interferometer is only the method to directly characterize the coherence properties of moiré exciton system.

We have added the descriptions in the main text (page 5).

2. Following on the use of nanostructure fabrication to achieve emission from a single localized interlayer exciton, do the authors have any statistics regarding the frequency with which they obtain a single localized interlayer exciton per confocal optical spot? It seems to me that the PL spectra measured at the pillar used in the first part of the work also shows PL from at least 2 different excitons (~1.38 and ~1.39 eV) even at the lowest excitation shown in the paper (although with the exciton at ~1.38 eV being considerably brighter than the other one).

We thank the Reviewer for helpful comments. As pointed out by the Reviewer, the statistical data provide the readers with important information. Figure R5 shows the statistical data of PL spectra from various nano-pillars with a size of 50 nm. A single peak and a few peaks are observed from a nano-pillar in the spectrum due to the statistical variations. However, the most of the spectra show single peak or major peak with dominant intensities. Thus, we can realize to observe an isolated and single PL peak in some nano-pillars with the size of 50 nm to be able to determine the coherence time by the first-order correlation functions.

Figure R8 (a)-(e) PL spectra of the nanopillars with diameter of 50 nm, measured at various nanopillars positions. (f) Optical image in the nano-fabricated heterobilayer. The red squares indicate the measured positions in the spectra.

3. Throughout the work, the authors claim the moiré superlattice to be responsible for the spatial confinement of their interlayer excitons. What evidence do the authors have for this? I note that, contrary to previous works on moiré-trapped interlayer excitons in MoSe₂/WSe₂, the authors do not present any evidence about the selection rules nor the magnetic properties characteristic of moiré-trapped interlayer excitons (see Nature 567, 66 (2019)). Moreover, for the twist angle estimated from the optical images (~1°), one should expect a strong lattice reconstruction for both 2H- and 3R-stacked MoSe₂/WSe₂ heterobilayers. Do the authors know their relative stacking orientation (i.e., 2H vs 3R)? Also, it's worth noting that for nearly lattice-matched 2D heterobilayers such as MoSe₂/WSe₂ there are theoretical predictions of exciton localization in the hot spots between a network of domain walls separating domains of commensurate stacking (see npj 2D Mater. Appl. 6, 74 (2022)). For all these reasons, I think it would be more appropriate to refer to the excitons in this work as “spatially localized interlayer excitons” rather than “moiré excitons”, since the authors do not prove their moiré origin.

Thank you for your helpful comments. This question is strongly related to Q3 from Reviewer #2. The additional data of SHG and polarization-resolved PL measurements suggest that the stacking-type and -angle of twisted semiconducting heterobilayer is H-type stacking and 56.5 ± 0.3 degree, respectively.

As pointed out by the Reviewer, there is some possibilities of the exciton localization in the hot spots (spatially localized interlayer exciton) in the commensurate stacking heterobilayer of MoSe₂/WSe₂. However, the value of gap modulation from net of domain walls decreases as the twist angle increase, as demonstrated in the paper (*npj 2D Mater. Appl.* 6, 74 (2022)). The rather shallow quantum dots arrays in antiparallel (AP)-bilayer with twist angle around 3.5 degree are unable to explain the energy range of sharp emission lines in our experiment.

With judging from the result of twist angle and polarization experiment, we think that the experimentally observed PL signals comes from the moiré excitons in the semiconducting heterobilayer.

4. In the interferograms measured for different temperatures (or exciton densities), the periodicity of the interference fringes does not seem to evolve monotonically with the temperature (or excitation power). For example, in Suppl. Fig 9, the periodicity of the fringes seems to be longer for 80 W/cm² than for 14 W/cm², but it seems to decrease again at 200 W/cm². A similar behavior is observed in Suppl. Fig. 10. Could the authors explain the origin of this?

Thank you for your helpful comments. The periodicity of the interference fringes is determined by the wavelength of the emission signal. In ideal case, the periodicities of the fringe in each interferogram should be exactly same, considering the same wavelength of their spectra. In actual measurement, the measurement conditions of data point numbers depending on the signal intensity and initial phase of the interference could lead to slightly different interferogram. However, such a small difference will not

influence the measurement results of coherence time, determined by whole decay profile of the visibility.

5. How do the authors estimate the visibility of the interferences in the interferograms as a function of delay time? The amplitude of the maxima and minima also changes with the delay time. Do the authors use consecutive maxima and minima? Or do they average within a given time window? I think the authors could add a sentence explaining how this is done.

Thank you for the helpful comments. In order to calculate the visibility in the interferogram, we use maxima and minima of the intensity within a given time window. At certain delay time, the maxima and minima intensities are determined within the range around an oscillation period, which can well present the conditions of constructive (in-phase) or destructive (anti-phase) interference pattern (update #16).

6. In the TRPL results, the authors explain that the measured curves follow a tri-exponential decay, from which they extract the three relevant decay times. These decay times go from 14 ns up to 700 ns at 4K. Then, the authors estimate an “average lifetime for the moiré exciton”. Can the authors explain how they estimate such an average time and what it’s its physical meaning?

The average lifetime is estimated by the equation: $\tau = (A_1\tau_1 + A_2\tau_2 + A_3\tau_3) / (A_1 + A_2 + A_3)$, where $A_{1,2,3}$ and $\tau_{1,2,3}$ are exacted from the tri-exponential fitting results. We have used this average lifetime to represent the lifetime of moiré exciton in the manuscript. However, after further consideration, we attributed the fastest decay component τ_1 to the biexciton lifetime, τ_2 to the moiré exciton lifetime, and τ_3 to the other slow decay component from the studies. We have revised the descriptions in the manuscript.

7. On a related comment, I think the authors should comment on which lifetime T_1 is being used in equation (3) to estimate the pure dephasing rate (and why).

Thank you for the helpful comments. The decay lifetime (τ_2) of moiré exciton is used as T_1 in the equation (3) for estimation of the pure dephasing time. Considering that time-distribution of the quantum two-level system is one of the decoherence mechanism, we should use the intrinsic decay lifetime of the system as the T_1 in equation (3) (update #18).

8. Finally, I think the authors could have done a better job referencing relevant previous work. It is surprising that, despite the authors’ claim of the moiré lattice being responsible for the interlayer exciton trapping, they do not cite the pioneering work from Xiaodong Xu’s lab on the first observation of moiré-trapped interlayer excitons (Nature 567, 66 (2019)). In a similar way, it is also surprising the omission of references to other previous works on moiré excitons, such as the demonstration of their single-photon emitter (quantum) nature (Sci. Adv. 6, eaba8526 (2020)), or their evolution from the single-quantum emitter to the interlayer exciton ensemble regime (Phys. Rev. X 11 (3), 031033 (2021)), among many others.

We thank the Reviewer for the important and helpful suggestion. According to the Reviewer's advice, we have added the important references to understand this study, and made revision in the manuscript.

Finally, below are some additional comments:

1. I believe the order (labelling) of supplementary figures S9 and S10 is reversed with respect to the description in the main text.

We thank the Reviewer for the comment. The correction has been made in the main text, and supporting information.

2. The authors state that the narrowest emission linewidth (~600 micro eV) is almost limited by the spectral resolution of the system. However, from panel Fig. 2c I can see that the narrowest emission line contains at least ~9 data points, which suggests to me that the PL lines are far from being resolution limited. This can also be seen in the noise of the PL background, where it's obvious than the resolution is better than 600 micro eV.

Thank you for your helpful comment. We evaluate the spectral resolution by the standard mercury lamp. The estimated linewidth of 600 μeV is calculated from the mercury emission line around the same energy position, which is affected by the slit width, CCD pixel size, color abbreviation and resolution of the monochromator. This value is determined under our experimental condition, which is considered as spectral resolution under our experimental condition. We have made revision in the main text (update #20).

Base on the helpful comments from the Reviewers, we added a more detailed discussion to our paper and we hope that we have now produced a better account for our work. We would like to express our appreciation to the Reviewers for their suggestions in improving our paper. We will be most grateful if you could offer us a second opportunity to review to *Nature Communications* with the revisions made. Look forward to hearing from you soon.

REVIEWERS' COMMENTS

Reviewer #1 (Remarks to the Author):

The authors have addressed most of the reviewers' concerns in the revised version of the manuscript and this has made the manuscript stronger. Following the reviewers' comments, the authors did the following critical experiments.

1. The authors fabricated reverse stacking in heterobilayers to demonstrate that the phenomenon is independent of the stacking order.
2. Additional SHG experiments were conducted for precise determination of twist angle and stacking order.
3. Linear and circular polarized PL as a function of excitation power for providing experimental evidence of moire exciton.

The two major issues raised by the reviewers in the previous version (a) Accurate determination of the twist angle (b) Whether the interlayer excitons are moire excitons have been addressed with the new experimental data.

Further, the authors have clarified how the current work is different from previous work on isolation and manipulation interlayer excitons. They have also shed light on why they used a different fabrication scheme (RIE over Ga⁺ ions) for the fabrication of nanopillars. With the new experimental data, more elaborate analysis and clarifications, I feel the current version of the manuscript is suitable for publication in Nature Communications.

Reviewer #2 (Remarks to the Author):

In this resubmission, Haonan Wang and his coauthors have diligently addressed and incorporated all the suggestions raised during the previous submission. The experimental approach, data manipulation, and analysis are all commendable and appropriate.

The authors have also provided a comprehensive description and citation of previous works related to the isolation and manipulation of interlayer excitons, thereby contextualizing and highlighting the state of the art within this manuscript. Additionally, they have included supplementary measurements such as polarized resolved PL and SHG experiments, which serve to further characterize the samples. These additional experiments aid in identifying the stacking angle and provide compelling evidence of the moiré nature of the PL emission. These were crucial aspects that warranted attention, and the authors have addressed them effectively.

While I consider that the study may represent an incremental advance in specific knowledge about interlayer excitons, this manuscript nevertheless stands as a robust contribution to a significant area of research. Based on this, I would recommend this work for publication in Nature Communications.

Reviewer #3 (Remarks to the Author):

After reading the authors' rebuttal letter, and the updated versions of the main and supplementary text (with all the additional experimental work carried out by the authors), I believe that the authors have done a very detailed and complete response to all the questions and comments I raised in my first review. I have also read through the authors' rebuttals to the points raised by the other reviewers, and in my opinion, they have satisfactorily answered many of them. The results of Wang et al. represent a valuable contribution to the community of researchers working in TMD heterostructures. Therefore, I think that the work is suitable for publication in Nature Communications in its current form.

RESPONSE TO REVIEWERS' COMMENTS

Reply to the comments of Reviewer #1

The authors have addressed most of the reviewers' concerns in the revised version of the manuscript and this has made the manuscript stronger. Following the reviewers' comments, the authors did the following critical experiments.

- 1. The authors fabricated reverse stacking in heterobilayers to demonstrate that the phenomenon is independent of the stacking order.**
- 2. Additional SHG experiments were conducted for precise determination of twist angle and stacking order.**
- 3. Linear and circular polarized PL as a function of excitation power for providing experimental evidence of moire exciton.**

The two major issues raised by the reviewers in the previous version (a) Accurate determination of the twist angle (b) Whether the interlayer excitons are moire excitons have been addressed with the new experimental data.

Further, the authors have clarified how the current work is different from previous work on isolation and manipulation interlayer excitons. They have also shed light on why they used a different fabrication scheme (RIE over Ga⁺ ions) for the fabrication of nanopillars. With the new experimental data, more elaborate analysis and clarifications, I feel the current version of the manuscript is suitable for publication in Nature Communications.

We thank the reviewer for the summary of our work and the recommendation for publication.

Reply to the comments of Reviewer #2

In this resubmission, Haonan Wang and his coauthors have diligently addressed and incorporated all the suggestions raised during the previous submission. The experimental approach, data manipulation, and analysis are all commendable and appropriate. The authors have also provided a comprehensive description and citation of previous works related to the isolation and manipulation of interlayer excitons, thereby contextualizing and highlighting the state of the art within this manuscript. Additionally, they have included supplementary measurements such as polarized resolved PL and SHG experiments, which serve to further characterize the samples. These additional experiments aid in identifying the stacking angle and provide compelling evidence of the moiré nature of the PL emission. These were crucial aspects that warranted attention, and the authors have addressed them effectively. While I consider that the study may represent an incremental advance in specific knowledge about interlayer excitons, this manuscript nevertheless stands as a robust contribution to a significant area of research. Based on this, I would recommend this work for publication in Nature Communications.

We thank the reviewer for the recognition of our work and the recommendation for publication.

Reply to the comments of Reviewer #3

After reading the authors' rebuttal letter, and the updated versions of the main and supplementary text (with all the additional experimental work carried out by the authors), I believe that the authors have done a very detailed and complete response to all the questions and comments I raised in my first review. I have also read through the authors' rebuttals to the points raised by the other reviewers, and in my opinion, they have satisfactorily answered many of them. The results of Wang et al. represent a valuable contribution to the community of researchers working in TMD heterostructures. Therefore, I think that the work is suitable for publication in Nature Communications in its current form.

We thank the reviewer for the recognition of our work and the recommendation for publication.